# IR3D-Bench: Evaluating Vision-Language Model Scene Understanding as Agentic Inverse Rendering

**Hengyu Liu**[1,*] **Chenxin Li**[1,*]**, Zhengxin Li**[2]**, Yipeng Wu**[2]**, Wuyang Li**[3]**,**

**Zhiqin Yang**[1]**, Zhenyuan Zhang**[4]**, Yunlong Lin**[5]**, Sirui Han**[4,†]**, Brandon Y. Feng**[6,†]

[1]CUHK, [2]TJU, [3]EPFL, [4]HKUST, [5]XMU, [6]MIT

https://ir3d-bench.github.io/

## Abstract

Vision-language models (VLMs) excel at descriptive tasks, but whether they truly understand scenes from visual observations remains uncertain. We introduce IR3D-Bench, a benchmark challenging VLMs to demonstrate understanding through active creation rather than passive recognition. Grounded in the analysis-by-synthesis paradigm, IR3D-Bench tasks Vision-Language Agents (VLAs) with actively using programming and rendering tools to recreate the underlying 3D structure of an input image, achieving agentic inverse rendering through tool use. This "understanding-by-creating" approach probes the tool-using generative capacity of VLAs, moving beyond the descriptive or conversational capacity measured by traditional scene understanding benchmarks. We provide a comprehensive suite of metrics to evaluate geometric accuracy, spatial relations, appearance attributes, and overall plausibility. Initial experiments on agentic inverse rendering powered by various state-of-the-art VLMs highlight current limitations, particularly in visual precision rather than basic tool usage. IR3D-Bench, including data and evaluation protocols, is released to facilitate systematic study and development of tool-using VLAs towards genuine scene understanding by creating.

## 1 Introduction

*What I cannot create, I do not understand. —Richard Feynman*

Vision-language models (VLMs) have made striking progress on tasks that resemble surface-level scene comprehension: answering questions, describing layout, grounding text in pixels [1, 2, 3, 4, 5, 6]. However, understanding, in a deeper sense, remains questionable. When a VLM generates text tokens that convey the meaning of three red cylinders in a scene, does its internal world model actually know what this means? Can it prove true understanding by reconstructing the world that image came from?

This paper proposes a different test of visual intelligence, one grounded not in passive recognition but in active creation through tool use. Inspired by the analysis-by-synthesis paradigm in human perception [7, 8, 9, 10], we frame this challenge as **agentic inverse rendering**: a vision-language agent (VLA) – an agent powered by a VLM – reconstructing the 3D scene behind a single image by writing an explicit, executable program that recreates the scene from scratch.

Agentic inverse rendering embodies the cyclical process of analysis-by-synthesis: analyzing the input, synthesizing a hypothesis, comparing it to the original, and refining based on the comparison. Similar to recent progress in tool-augmented VLMs [11, 12, 13, 14], our programmatic approach recasts the traditional goal of inverse rendering as a multimodal programming task where the VLA must

---

*Equal Contribution, † Corresponding Author.

39th Conference on Neural Information Processing Systems (NeurIPS 2025) Track on Datasets and Benchmarks.

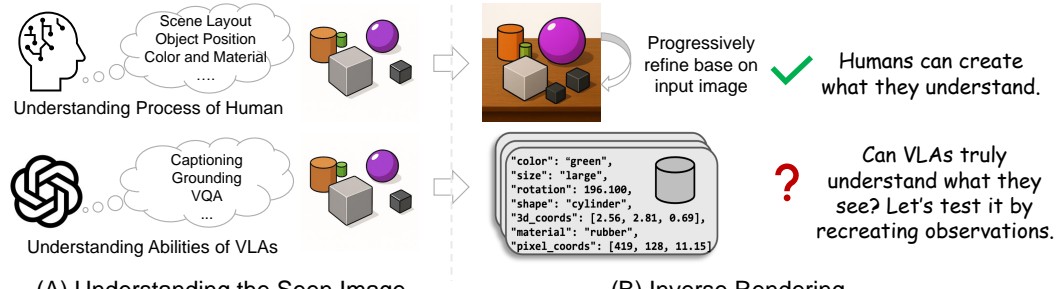

(A) Understanding the Seen Image     (B) Inverse Rendering

Figure 1: Understanding by Creating. (A) Humans demonstrate understanding by constructing internal world models with mental representations of object layouts, spatial relations, and physical attributes, which they can use to recreate observed scenes. In contrast, vision–language agents (VLAs) are typically evaluated on recognition tasks like captioning or VQA, which only tap into surface-level visual comprehension. (B) IR3D-Bench shifts the focus to generative reconstruction with inverse rendering via tool use. While VLAs show sparks of scene understanding, their ability to build coherent, executable 3D programs reveals how incomplete their internal world models remain.

actively use tools to demonstrate understanding: to recreate a given image, the agent must compose a valid Python script, communicated to Blender [15] via Model Context Protocol (MCP) [16, 17], that expresses the underlying 3D structure when executed by Blender. This tool-using programmatic representation lifts the VLA's internal world model beyond vectorized features to an explicit 3D physical space. However, despite the early promise shown by VLA using tools like BlenderMCP [17] to generate 3D content based on 2D image prompts, the generated outputs remain unreliable. More importantly, the limitations in scene understanding of current VLM are not yet well understood.

To systematically evaluate VLM scene understanding through the lens of "understanding-by-creating", we introduce the **IR3D-Bench** benchmark, where we prompt the VLA with an image and task it to perform agentic inverse rendering by producing a Blender script that, when executed, reconstructs the original scene. This explicitly tests the agent's ability to use programming tools to externalize its understanding. We evaluate VLA performance using a suite of metrics that assess geometry, spatial relations, appearance attributes, and overall plausibility.

Unlike existing multimodal benchmarks that primarily focus on descriptive or conversational tasks such as 3D captioning, 3D Visual Question Answering (VQA), or 3D visual grounding, IR3D-Bench directly tests the agentic generative capacity: the ability to use tools to synthesize the latent 3D world state from a 2D view. Our experiments suggest that while scripting errors occur, the dominant bottleneck is not tool usage itself, but a lack of visual precision in the agent's perception. When agents cannot reliably distinguish fine-grained differences between their rendered output and the target image, they quickly plateau in self-correction, even with iterative prompting. This indicates that future progress may hinge less on instruction tuning or syntax scaffolding and more on enhancing the fidelity of visual representations within multimodal models.

We release IR3D-Bench to facilitate the systematic study of VLM scene understanding and the development of agents that can observe, reason about, and truly understand scenes by demonstrating the ability to recreate them in an actionable, structured format. Just as human children show early signs of understanding by attempting to recreate what they see [18, 19, 20], our benchmark embraces Feynman's insight that creation is the test of understanding and challenges AI systems to prove their comprehension by generating, through tool use, the world they perceive [21, 22].

## 2 Agentic Inverse Rendering with VLMs

This section formally defines our task of agentic inverse rendering. We discuss connections to the broader field of inverse rendering, while highlighting the fundamental shift enabled by VLAs.

Traditional inverse rendering seeks to infer the underlying scene properties (*e.g.,* geometry, appearance, illumination) that best explain the observation. This is often framed as an optimization problem

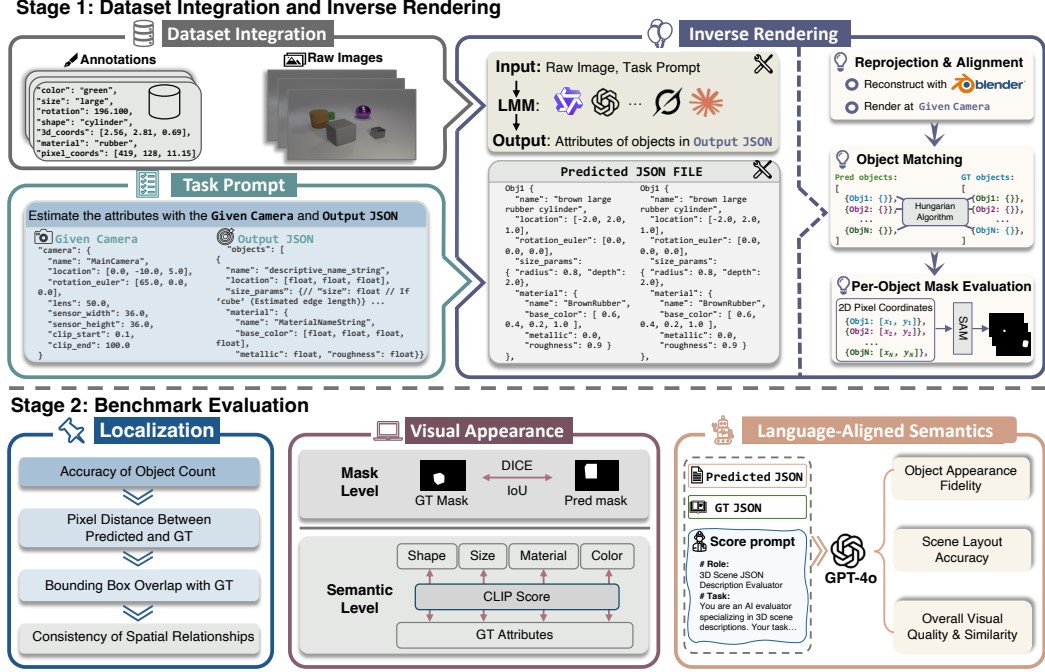

Figure 2: Overview of the IR3D-Bench Pipeline. The benchmark consists of two stages: Stage 1: Inverse Rendering. Given a raw image and camera parameters, the agent is prompted to infer a structured scene representation in JSON format. The predicted objects are rendered in Blender and matched to GT annotations using geometric alignment and per-object mask comparisons. Stage 2: Benchmark Evaluation. Reconstruction quality is evaluated along three axes: Localization (object count, spatial alignment, and relation consistency), Visual Appearance (shape and material accuracy via mask- and attribute-level scores), and Language-Aligned Semantics (layout fidelity and object plausibility assessed via GPT-4o). Together, these metrics provide a comprehensive view of the VLA's internal world model and generative precision.

over a differentiable graphics pipeline: Given a rendering model and camera parameters, the goal is to find 3D scene parameters that minimize the difference between the rendered image and the input.

Our task differs substantially from conventional approaches to 3D inverse rendering through a focus on agentic tool use. Rather than performing low-level predictions like geometric primitives, depth maps, or volumetric densities, the VLA actively uses programming tools to synthesize an explicit, executable program that regenerates the 3D scene depicted in a single input image $\mathbf{I}$. This program $\mathcal{P}$, typically a Python script for Blender, acts as a structured, symbolic hypothesis about the scene composition. When executed within a deterministic rendering environment $\mathcal{R}$, the script should produce an image $\mathcal{R}(\mathcal{P})$ that matches the original $\mathbf{I}$ as closely as possible under a set of predefined metrics. This directly embodies the analysis-by-synthesis cycle: the VLA analyzes $\mathbf{I}$ and synthesizes $\mathcal{P}$ as its understanding of the scene's generative process.

This shift from continuous parameter regression (classic inverse rendering) to structured program synthesis introduces distinct challenges. The VLA must use programming tools effectively, producing outputs that are not only semantically meaningful but also syntactically precise. A syntactically invalid script is unexecutable; subtle errors in coordinate systems can lead to geometrically plausible but incorrect reconstructions; and inconsistencies in naming conventions can cause API errors. These strict requirements of tool use limit the tolerance for hallucinations, imprecision, or vagueness often seen in VLM outputs for free-form tasks like captioning or dialogue, and demand a higher level of precision in the agent's understanding of both the visual scene and the programming tools.

To manage the complexity of unconstrained 3D scenes and to enable rigorous evaluation, our study constrains the task domain using scenes from the CLEVR dataset [23]. CLEVR provides ground-truth scene graphs, 6D object poses, material properties, and other metadata for scenes composed of simple

primitive objects under controlled lighting. This controlled environment allows us to focus evaluation on the agent's ability to use programming tools effectively rather than on scene complexity itself.

The interface with the 3D environment (*e.g.,* Blender MCP) is crucial for agentic inverse rendering. It acts as the bridge, interpreting the VLA's generated text as executable code. This programmatic formulation allows us to assess model performance at multiple levels of abstraction: Does it correctly place objects in 3D space (geometric accuracy)? Does it preserve spatial relationships between objects (relational reasoning)? Does it match visual features like color, material, and shape (appearance fidelity)? Is the overall reconstructed scene plausible and similar to the original (holistic quality)?

These evaluations probe not only the VLA's ability to act as an agent using Blender tools but its capacity to internalize and reconstruct a latent 3D world state from a 2D image. IR3D-Bench thus positions agentic inverse rendering not merely as a reconstruction task, but as a foundational step towards agents that possess a practical, generative understanding of the 3D world, enabling them to not just perceive it, but also to reconstruct and manipulate it.

# 3   Related Work

Our work intersects with several established and emerging research areas: classic inverse rendering, broader efforts in 3D scene understanding using VLMs, and programmatic scene modeling.

**Classic Inverse Rendering** Traditional inverse rendering has long sought to recover the underlying physical and geometric properties of a scene, such as shape, material, and illumination, from one or more 2D images [24]. This is typically formulated as an optimization problem where the parameters of a (often differentiable) graphics pipeline are adjusted to minimize the discrepancy between a rendered image and the observed input [25, 26, 27]. This paradigm has led to substantial progress in physically grounded scene reconstruction, with notable advances in volumetric or implicit neural representations [28, 29, 30], point-based rendering [31, 32] and differentiable path tracing [33]. These methods can achieve high-fidelity reconstructions and are often grounded in physical priors. Our approach, while also a form of inverse rendering, shifts the target representation from continuous weights or geometric primitives to discrete, executable programs, prioritizing interpretability and editability, and explicit demonstration of understanding through creation.

**VLMs for 3D Scene Understanding** The capabilities of Vision-Language Models (VLMs) have recently been extended to the 3D domain [34, 35]., leading to benchmarks and methods for tasks such as 3D Visual Question Answering [36, 37], 3D visual grounding [38, 39], 3D captioning [40, 41], and embodied AI navigation or interaction based on language instructions [42, 43]. These works primarily focus on the VLM's ability to interpret or describe existing 3D information, whether presented as point clouds, meshes, or within simulated environments [44, 45, 46]. Some recent efforts also explore 3D-aware LLMs or multimodal instruction tuning for broader 3D reasoning [47, 48, 49, 50, 51, 52]. While these approaches advance 3D scene comprehension, they generally do not position the model as an agent using programming tools to generate the underlying 3D scene structure from a single 2D image in an explicit, programmatic form. Our work uniquely focuses on this agentic, tool-using generative synthesis capability as a test of deeper understanding demonstrated through creation.

**Programmatic Scene Generation and Analysis-by-Synthesis** Representing scenes as programs or through generative grammars has a rich history in computer graphics and vision [53, 54, 55, 56]. More recently, with the advent of powerful tool-augmented language models, there is renewed interest in having models generate code that produces complex outputs [11], including visual content [57, 58, 59, 60, 61, 62, 63]. This echoes with the analysis-by-synthesis paradigm [8], where perception involves generating internal hypotheses (programs) to explain visual sensory input. Some works generate 2D images or simple 3D assets programmatically from text or sketches [58, 64, 65]. Others focus on programs that control simulation environments [66] or robotic actions [67]. However, the specific task of a VLA reconstructing a detailed 3D scene from a single RGB image by using programming and rendering tools, and systematically benchmarking this agentic "understanding-by-creating" ability, remains less explored, and IR3D-Bench directly addresses this gap.

# 4 IR3D-Bench Suite

In this section, we introduce the core components of IR3D-Bench, our proposed benchmark for evaluating VLMs scene understanding via agentic inverse rendering shown in Fig. 2. Sec. 4.1 describes how we integrate the CLEVR dataset for our benchmark. Sec. 4.2 presents the inverse rendering pipeline, which involves VLAs using tools to reconstruct 3D scene representations from visual inputs. Sec. 4.3 defines a set of evaluation metrics to assess VLA performance.

## 4.1 CLEVR Dataset Integration

CLEVR dataset [23] has been widely adopted in 3D vision tasks [68, 69, 70, 71, 72]. To facilitate controlled evaluation of agentic inverse rendering and 3D scene understanding, we adopt the validation split of CLEVR, which contains 15,000 synthetic images rendered from 3D scene graphs. Each image depicts a structured scene composed of 3 to 10 objects, with precise annotations of object-level geometry and semantics, including 3D coordinates $\mathbf{P}_{3D} \in \mathbb{R}^3$, pixel-space projections $\mathbf{P}_{2D} \in \mathbb{R}^3$, shapes $\mathbf{t}_s$, colors $\mathbf{t}_c$, sizes $\mathbf{t}_z$, materials $\mathbf{t}_m$, and inter-object spatial relationships defined as: for each spatial relation $r \in \{\text{right}, \text{left}, \text{front}, \text{behind}\}$, $\mathbf{R}_i^{(r)} \subseteq \{0, 1, \ldots, N-1\} \setminus \{i\}$, where $\mathbf{R}_i^{(r)}$ denotes the set of objects that are in relation $r$ with object $i$ and $N$ be the number of objects, indexed by $i = 0, 1, \ldots, N-1$. The images are rendered at a resolution of $480 \times 320$ pixels using Blender [15]. These rich annotations make CLEVR particularly suitable for benchmarking object-centric 3D reconstruction and spatial reasoning in a controlled setting.

In IR3D-Bench, we leverage CLEVR as a structured testbed for evaluating VLAs as tool-using agents. For each scene, we use the rendered RGB image from CLEVR along with a well-designed textual prompt that specifies the agentic inverse rendering task, and provide both as input to the VLA. The prompt is constructed under fixed camera intrinsic $\mathcal{K}$, and extrinsic $\mathcal{E}$ across all samples, and encodes the reconstruction objective and relevant assumptions as illustrated in Fig. 7. The VLA is tasked with predicting a set of object-level parameters structured according to a predefined JSON schema.

## 4.2 Inverse Rendering Pipeline

We evaluate how effectively the agent has reconstructed the full 3D scene using Blender [15]. Here, we define the attributes from four dimensions that the agent must correctly specify: 3D position $\hat{\mathbf{P}}_{3D}$, shape $\hat{\mathbf{t}}_s$, color $\hat{\mathbf{t}}_c$, size $\hat{\mathbf{t}}_z$, and material $\hat{\mathbf{t}}_x$. Rendering is performed under a fixed camera intrinsic matrix $\mathcal{K}$ and extrinsic parameters $\mathcal{E}$, determining the rendering viewpoint. This controlled setup ensures consistent projection geometry across all evaluations.

**Reprojection and Alignment** The coordinate systems of the generated and the ground-truth (GT) scenes are misaligned in CLEVR, potentially leading to unfair or failed evaluation. To address this issue, we adopt a fixed camera model with known intrinsics $\mathcal{K}$ and extrinsics $\mathcal{E}$ to project the predicted 3D object centers into the 2D image plane. The projection is defined as:

$$\hat{\mathbf{p}}_{2D} = \pi(\mathcal{K}, \mathcal{E}, \mathbf{P}_{3D}) = \mathcal{K} \cdot [\mathcal{R} \mid \mathbf{t}] \cdot \begin{bmatrix} \mathbf{P}_{3D} \\ 1 \end{bmatrix},$$

where $\hat{\mathbf{P}}_{3D} \in \mathbb{R}^3$ is the predicted object center in world coordinates, and $\hat{\mathbf{p}}_{2D}$ denotes its corresponding location in the image space.

**Object Matching** To resolve object correspondences between agent-generated reconstructions (Fig. 2 Stage 1) and GT, we first augment each agent-generated object with a textual *description name* (e.g., "red large metal sphere") that encodes its semantic attributes: $\hat{\mathbf{t}}_c + \hat{\mathbf{t}}_z + \hat{\mathbf{t}}_m + \hat{\mathbf{t}}_s$. We then extract structured attribute vectors from both agent-generated and GT objects, denoted as $\hat{\mathbf{t}}_i$ and $\mathbf{t}_j$ respectively, where $\hat{\mathbf{t}}_i$ represents the attribute set (color, size, shape, material) of the $i$-th predicted object, and $\mathbf{t}_j$ that of the $j$-th GT object. For each attribute dimension $k$ in color, size, material, and shape, we compute the semantic similarity using a CLIP [73] text encoder: $s(\hat{\mathbf{t}}_i, \mathbf{t}_j) = \frac{1}{4} \sum_{k=1}^{4} \text{CLIP}\left(\hat{\mathbf{t}}_i^{(k)}, \mathbf{t}_j^{(k)}\right)$, where $\hat{\mathbf{t}}_i^{(k)}$ and $\mathbf{t}_j^{(k)}$ denote the $k$-th attribute (as a text string) of the $i$-th predicted and $j$-th GT object. This yields a similarity matrix $S \in \mathbb{R}^{N \times M}$ between the $N$ predicted and the $M$ GT objects. We convert it into a cost matrix $C = 1 - S$, and solve the assignment using the Hungarian algorithm: $\phi^* = \arg\min_\phi \sum_{i=1}^{N} C_{i,\phi(i)} = \arg\max_\phi \sum_{i=1}^{N} S_{i,\phi(i)}$,

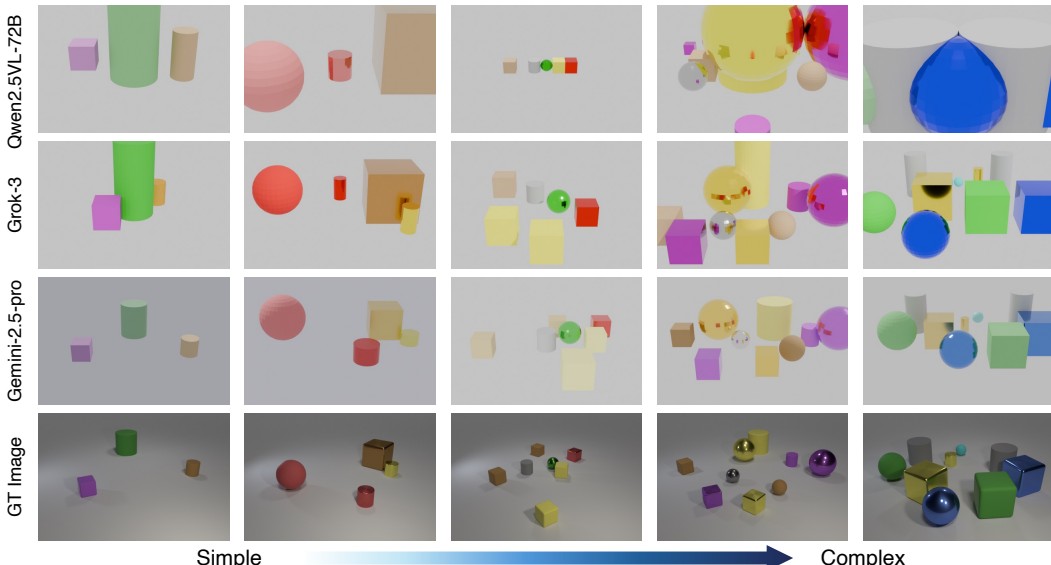

Figure 3: Visual Results with Selected VLMs. Gemini-2.5-pro demonstrates strong understanding of object spatial positions and relative layouts. Grok-3 excels at modeling fine-grained details such as material and color. Qwen2.5-VL-72B struggles in more complex scenarios.

where $\phi$ is a bijective mapping from the set of predicted object indices $\{1, \ldots, N\}$ to GT object indices $\{1, \ldots, M\}$. To avoid spurious matches, we apply a similarity threshold $\tau$ and define the final matching function as $\text{Match}(i) = \phi^*(i)$ if $S_{i,\phi^*(i)} \geq \tau$, and $\text{Match}(i) = -1$ if otherwise.

**Per-Object Mask Evaluation** To quantitatively assess the fidelity of each reconstructed object, we employ the Segment Anything Model (SAM) [74] to extract instance-level segmentation masks in a zero-shot manner. Specifically, we project the 3D center of each predicted object onto the image plane to obtain a 2D point $\hat{\mathbf{p}}_{2D}$, which serves as the prompt to SAM. The model then returns a binary segmentation mask $\hat{\mathbf{M}}_i$ corresponding to the predicted object $i$. Similarly, we apply the same procedure to the GT 3D object centers to extract ground-truth masks $\mathbf{M}_j$ via the same SAM-based pipeline, ensuring consistent and unbiased mask generation.

## 4.3 Evaluation Metrics

To comprehensively evaluate the performance of VLMs on agentic inverse rendering and 3D scene understanding, we propose a suite of evaluation metrics grouped into four major categories: **Localization**, **Visual Appearance**, and **Language-Aligned Semantics**. This multi-faceted evaluation protocol allows us to assess both geometric accuracy and semantic fidelity of the reconstructed scenes.

### 4.3.1 Localization

We evaluate object-level localization from four perspectives: *geometric accuracy*, *object count consistency*, *bounding box similarity* and *spatial relations*. These metrics jointly assess how accurately the model reconstructs spatial layouts in the scene.

**Pixel Distance** We compute the average $\ell_2$ distance between the 2D projected centers of generated objects $\hat{\mathbf{p}}_i$ and their corresponding GT centers $\mathbf{p}_j$, after optimal bipartite matching. This quantifies the geometric proximity between generated and GT object positions in image space.

**Count Accuracy** We evaluate the consistency of object enumeration by comparing the generated object count with the GT number of objects.

**BBox Edge Score** To quantify the structural similarity between generated and GT bounding boxes, we propose a center-to-edge distance-based metric. Given a predicted box $\hat{b}_i = (x_1, y_1, x_2, y_2)$ and the GT box $b_j$, we first compute the distances from the box center to each of its four edges. These

directional distances capture both the size and aspect ratio of the box. BBox Edge Score is then formulated as $s_{\text{bbox}}(\hat{b}_i, b_j) = 1 - \frac{\sum_k \left| d_k^{(i)} - d_k^{(j)} \right|}{\sum_k |d_k^{(j)}| + \epsilon}$, where $d_k^{(i)}$ and $d_k^{(j)}$ represent the center-to-edge distances for the generated and GT boxes along each direction $k \in \{\text{left, right, top, bottom}\}$, and $\epsilon$ is added to prevent division by zero. This metric yields a score in the range $[0, 1]$, where higher values indicate greater similarity in both size and alignment.

**Spatial Relations** We measure how well the VLA recovers pairwise object relations like `left of`, `right of`, `in front of`, and `behind`. Given the predicted 3D positions $\hat{\mathbf{P}}_{3D}$, we derive relational labels based on predefined geometric rules and compare them with GT $\{\mathbf{R}_i^{(r)}\}$. Relation Accuracy is reported as the proportion of correctly predicted relations across all annotated object pairs.

### 4.3.2 Visual Appearance

**Mask-level** We evaluate the quality of generated segmentation masks $\hat{M}_i$ obtained in Per-Object Mask Evaluation by comparing them with GT masks $M_j$ using Intersection-over-Union (IoU) and DICE Score. These metrics capture spatial overlap and foreground prediction accuracy, respectively.

**Semantic-level** To evaluate the consistency of object-level semantic attributes, we convert both predicted and GT properties (color, size, material, and shape) into textual descriptions. These are embedded using the CLIP model, and cosine similarity is computed between the predicted and reference embeddings. Attribute-wise scores are reported along with an Overall Appearance Score obtained by averaging across all annotated attributes.

### 4.3.3 Language-Aligned Semantics

We use GPT-4o to assess perceptual quality and semantic coherence as LLM score. Both predicted and GT scenes are JSON-serialized and presented to GPT-4o, which provides ratings from 0 to 5 in three dimensions: 1) *Object Appearance*, correctness of color, shape, and material; 2) *Scene Layout*, consistency of the spatial object arrangement with GT; 3) *Overall Visual Quality*, holistic realism and semantic alignment of the entire scene. The evaluation prompt used is shown in Figure 7 (A).

## 5 Experiments

In this section, we evaluate the agentic inverse rendering capabilities of VLMs as they interact with programming and rendering tools to demonstrate their understanding by creating.

### 5.1 Benchmark Setup

**Models** We evaluate more than 20 state-of-the-art VLMs, with diverse model sizes (from 2B to 70B), architecture, and training paradigms. Closed-source models include GPT-4o [2], GPT-4.1 [75], GPT-4V [75], Gemini-2.0-Flash [76], Gemini-2.5-Pro [76], Claude-3.5-Sonnet [3], Claude-3.7-Sonnet [3], and Grok-3 [77]. Open source models include DeepSeek-VL2 [78],

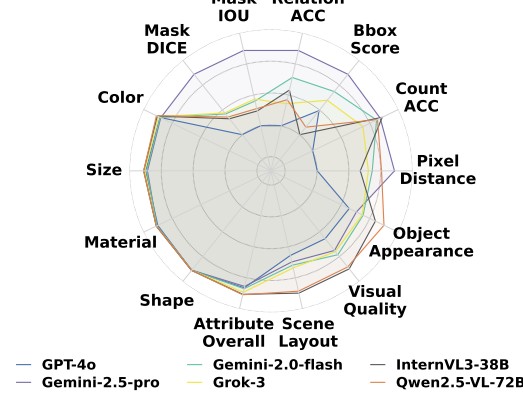

Figure 4: Holistic comparison over 14 metrics.

LLaVA-NeXT [79], LLaMA-3.2 [80], H2OVL [81], Phi-3.5-Vision [82], Pixtral [83], Aria [84], Idefics [85], InternVL2.5 [86], InternVL3 [87], and Qwen2.5-VL [88].

**Prompt Design** We design a single-turn prompt that instructs VLM to extract geometric information from a given image. The VLM predicts the attributes of each object, such as shape, size, position, and material, based on a predefined output format, returning a structured JSON file. The intrinsic and extrinsic parameters of the camera are fixed and cannot be inferred, ensuring a consistent reconstruction of the scene (see Sec. 5.2.3 for details). The full prompt is provided in the appendix.

Table 1: **Evaluation of VLMs on IR3D-Bench.** We report performance across key aspects of 3D scene reconstruction from a single image. For each column, only the top three performances are highlighted from dark (highest) to light (lowest).

| Model | Release | Layout & Localization | | | Relation | Instance Seg. | | CLIP Score | | | | | LLM Score | | |
|---|---|---|---|---|---|---|---|---|---|---|---|---|---|---|---|
| | | Pix. Dist.↓ | Count Acc↑ | Bbox↑ | Rel. Acc↑ | IoU↑ | Dice↑ | Color↑ | Size↑ | Material↑ | Shape↑ | Overall↑ | Obj App.↑ | Layout↑ | Overall↑ |
| **Latest Proprietary Models** | | | | | | | | | | | | | | | |
| Gemini-2.5-pro | 2025-03 | 0.3791 | 1.00 | 0.45 | 0.55 | 0.11 | 0.18 | 96.12 | 97.00 | 99.50 | 99.75 | 93.08 | 2.96 | 2.05 | 2.62 |
| Gemini-2.0-flash | 2025-02 | 0.4291 | 0.99 | 0.37 | 0.46 | 0.08 | 0.13 | 96.59 | 97.67 | 99.41 | 99.92 | 94.97 | 2.99 | 2.08 | 2.72 |
| Claude3.5-Sonnet | 2024-10 | 0.5402 | 0.87 | 0.50 | 0.28 | 0.09 | 0.14 | 93.19 | 96.77 | 97.39 | 98.60 | 91.39 | 2.67 | 1.85 | 2.28 |
| Claude-3.7-Sonnet | 2025-02 | 0.5099 | 0.93 | 0.53 | 0.38 | 0.09 | 0.14 | 97.71 | 98.34 | 99.42 | 99.09 | 96.36 | 3.05 | 2.10 | 2.82 |
| GPT-4.1 | 2025-04 | 0.4366 | 1.00 | 0.48 | 0.42 | 0.08 | 0.13 | 97.55 | 97.34 | 98.96 | 99.87 | 94.59 | 2.68 | 1.66 | 2.34 |
| GPT-4o | 2024-11 | 0.5528 | 0.94 | 0.29 | 0.30 | 0.07 | 0.11 | 96.70 | 98.36 | 98.66 | 99.88 | 94.22 | 2.90 | 1.94 | 2.52 |
| grok-3 | 2024-12 | 0.4378 | 0.98 | 0.33 | 0.38 | 0.08 | 0.13 | 98.04 | 99.15 | 99.87 | 99.89 | 97.80 | 3.02 | 2.06 | 2.71 |
| **Open-source Models** | | | | | | | | | | | | | | | |
| DeepSeek-VL2 | 2024-12 | | | | | | | | | | | | | | |
| Llama-3.2-11B-Vision | 2024-09 | | | | | | × Failed | | | | | | | | |
| H2OVL-Mississipi-2B | 2024-10 | | | | | | | | | | | | | | |
| LLaVA-NeXT | 2025-01 | 0.6835 | 0.69 | 0.38 | 0.12 | 0.03 | 0.04 | 92.11 | 96.78 | 96.31 | 96.85 | 89.17 | 2.03 | 0.96 | 1.47 |
| Mistral3 | 2025-01 | 0.4733 | 0.99 | 0.26 | 0.44 | 0.06 | 0.11 | 99.56 | 99.79 | 99.85 | 99.90 | 97.95 | 3.17 | 2.16 | 2.78 |
| Phi-3.5-Vision | 2024-07 | 0.6027 | 0.80 | 0.45 | 0.13 | 0.02 | 0.03 | 91.44 | 96.35 | 93.08 | 96.35 | 87.06 | 2.10 | 1.01 | 1.53 |
| phi4_mm | 2025-02 | 0.6192 | 0.92 | 0.32 | 0.21 | 0.03 | 0.05 | 94.82 | 93.16 | 96.02 | 99.58 | 92.63 | 2.59 | 1.49 | 2.04 |
| Pixtral-12B | 2024-11 | 0.4661 | 0.98 | 0.23 | 0.42 | 0.07 | 0.11 | 99.28 | 99.90 | 99.03 | 99.83 | 98.93 | 3.22 | 2.15 | 2.78 |
| Aria | 2024-11 | 0.5932 | 0.87 | 0.25 | 0.17 | 0.05 | 0.08 | 95.96 | 99.22 | 92.22 | 99.80 | 92.09 | 2.90 | 1.91 | 2.44 |
| Idefics3-8B | 2024-08 | 0.9100 | 0.97 | 0.11 | 0.18 | 0.03 | 0.06 | 98.35 | 99.83 | 95.35 | 99.98 | 96.97 | 3.14 | 1.79 | 2.48 |
| InternVL2.5-8B | 2024-11 | 0.9511 | 1.00 | 0.22 | 0.28 | 0.03 | 0.05 | 99.85 | 99.92 | 99.85 | 99.98 | 99.80 | 3.02 | 1.86 | 2.51 |
| InternVL2.5-38B | 2024-11 | 0.5233 | 1.00 | 0.23 | 0.38 | 0.07 | 0.11 | 99.79 | 99.98 | 100.00 | 100.00 | 99.86 | 3.26 | 2.17 | 2.83 |
| InternVL3-8B | 2025-04 | 0.5549 | 1.00 | 0.32 | 0.30 | 0.05 | 0.08 | 99.20 | 99.49 | 98.82 | 99.96 | 98.82 | 3.00 | 1.89 | 2.49 |
| InternVL3-38B | 2025-04 | 0.4560 | 1.00 | 0.18 | 0.42 | 0.07 | 0.13 | 99.15 | 99.98 | 100.00 | 100.00 | 99.47 | 3.25 | 2.22 | 2.89 |
| Qwen2.5-VL-7B | 2025-01 | 0.6537 | 0.96 | 0.40 | 0.30 | 0.04 | 0.06 | 98.21 | 99.60 | 99.71 | 99.86 | 96.89 | 3.04 | 1.95 | 2.55 |
| Qwen2.5-VL-72B | 2025-01 | 0.4082 | 1.00 | 0.21 | 0.39 | 0.08 | 0.13 | 99.86 | 99.98 | 99.99 | 99.98 | 99.80 | 3.24 | 2.20 | 3.02 |

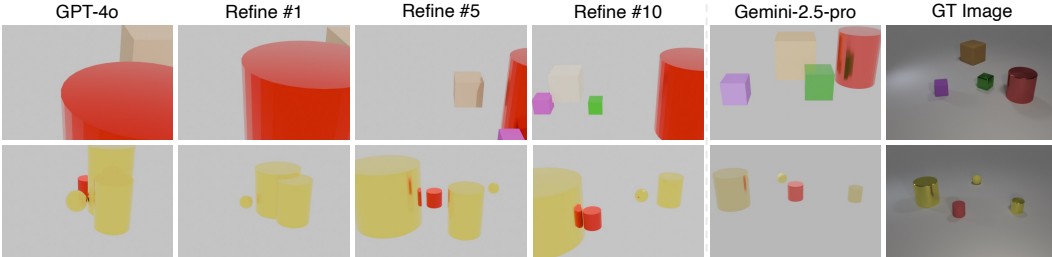

Figure 5: Qualitative results illustrating the effect of *increasing refinement iterations* on performance. Starting from GPT-4o outputs, iterative refinements (#1, #5, #10) progressively improve alignment with the GT. Gemini-2.5-pro results are also shown for comparison.

## 5.2 Experimental Results

**General Trends** As shown in Table 1, several models (DeepSeek-VL2 variants, LLaMA-3.2-11B-Vision, H2OVL-Mississippi-2B) failed to produce valid outputs, indicating either insufficient 3D understanding or incompatibility with the task format. Among those completing the benchmark, most VLMs demonstrate strong recognition of object-level attributes (color, material, shape, size) as indicated by high CLIP scores (GPT-4o: 0.98, Claude-3.7: 0.95); Pixel Distance is low (GPT-4o: 0.0004, Gemini-2.5-pro: 0.0003), showing most models can well estimate the position of the object center. However, their spatial understanding between objects is notably weaker. Size-related metrics such as IoU and DICE are low (IoU: GPT-4o: 0.43, Claude-3.7: 0.40), indicating difficulty in estimating object scale and boundaries; Relational Accuracy, which captures reasoning over inter-object spatial relations, is below 0.3 for most models (GPT-4o: 0.28, Gemini: 0.26), showing persistent errors in understanding relative positions, proximity, and containment. Together, these results suggest that while VLMs can describe what is in the scene and how it looks, they still struggle to understand where things are and how they relate in structured 3D space.

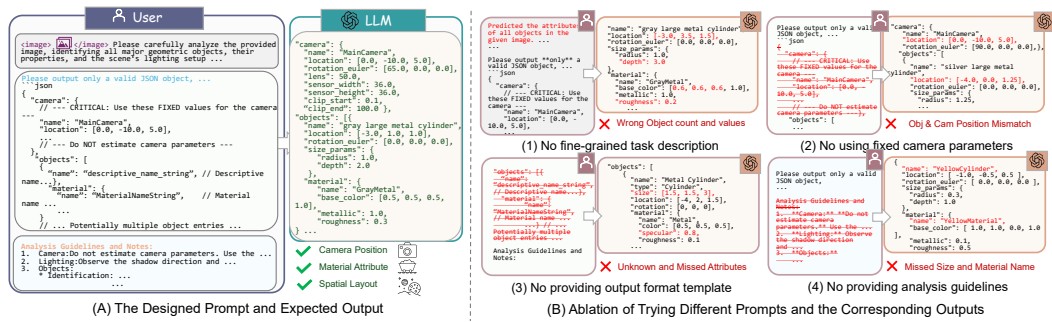

Figure 7: Prompt Design and Error Analysis. (A) Inverse rendering prompt includes structured format, fixed camera, and attribute guidelines. (B) Ablation of prompt components reveals distinct failure modes, highlighting the importance of precise prompt engineering.

**Model-specific Analysis** We conduct an in-depth analysis on two VLMs, Gemini-2.5-pro and Qwen-VL-2.5, which are representative of two paradigms (high-end proprietary model v.s. open-source model). Gemini-2.5-pro is consistently strong across all evaluated dimensions. With a CLIP score of 0.97, it shows precise recognition of object-level appearance (color, shape, and material). In terms of spatial reasoning, it achieves an IoU of 0.41 and a DICE score of 0.55, showing reliable estimation of object size and extent. Its Pixel Distance is low at 0.29, reflecting accurate object localization, while Relational Accuracy reaches 0.26, among the highest in our benchmark. These results suggest that Gemini-2.5-pro not only recognizes what is present in a scene, but also exhibits a measurable capacity to infer where objects are and how they are spatially organized. In contrast, Qwen-VL-2.5, as a leading open-source alternative, performs reasonably well in appearance-related tasks with a CLIP score of 0.89. However, its spatial understanding remains limited. The model records a Pixel Distance of 0.42 and low IoU and DICE scores of 0.28 and 0.36, which indicate difficulties in precise object localization and shape reconstruction. Its Relational Accuracy is 0.18, suggesting substantial challenges in modeling inter-object spatial relationships. Still, it shows consistent recognition of object categories and general scene layout, which points to a solid foundation for future improvements.

**Iterative Refinements** We examine if agentic inverse rendering can improve with iterative reasoning and self-correction, using GPT-4o as the backbone VLM (prompt design detailed in Appendix Tab. 4). At each refinement step, the VLA sees the GT image, the previous rendering, and the JSON scene description. We apply ten refinement iterations, and some renderings are shown in Figure 5. As the number of refinement steps increases, we observe consistent improvements in object spatial alignment, color fidelity, and scale accuracy. After ten iterations, the output quality becomes comparable to that of Gemini-2.5-pro. Figure 6 confirms this trend in quantitative metrics: pixel distance decreases while bounding box scores increase as we scale refinement iterations, suggesting that the model's scene understanding improves after refinement.

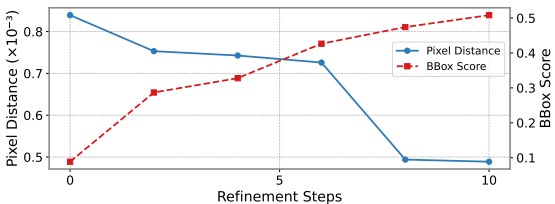

Figure 6: Understanding scales with more refinements.

**Impact of Prompt Design** We conduct ablation studies on four key components of our prompt: (1) task decomposition and clarification, (2) fixed camera parameters, (3) structured output format, and (4) detailed attribute guidelines. Figure 7 (B) shows that removing any component leads to failure cases. Task decomposition helps the model understand complex instructions and accurately determine object count. Fixed camera ensures consistent object orientation and spatial alignment. A structured output format guides the model to produce valid and complete JSON results. Clear attribute guidelines improve the accuracy of size, shape, and material predictions.

## 6 Conclusion

IR3D-Bench redefines VLM scene understanding through agentic inverse rendering, challenging VLAs to reconstruct 3D scenes from 2D images via automatic tool-use. Our experiments find that

current VLMs grasp high-level object attributes and tool-use abilities, but struggle with precise spatial control. We found that iterative refinement and careful prompt design can improve reconstruction quality, providing guidance for future VLM research. With IR3D-Bench, we provide the community with a systematic framework to measure progress of VLM scene understanding, moving beyond passive observation to agentic understanding-by-creating.

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

# A Experiment Setup Details

## A.1 Implement Details

**Setup**    To ensure consistency across models with reasoning capabilities, such as GPT-4o [89] and Grok-3 [77] etc, which often generate intermediate "thought" steps or chain-of-thought reasoning, we discard all non-structured outputs during inference. We extract only the final JSON-formatted response that conforms to our predefined schema. In designing the evaluation metrics, we intentionally ignore lighting-related attributes (shading and brightness) since these aspects are not explicitly modeled by the agents and often introduce high variability. Moreover, accurately recovering lighting conditions from a single image is inherently ambiguous, even for human annotators, making it an unrealistic expectation for current VLMs. Thus, we focus evaluation on geometry and semantics, rather than photometric fidelity. Additionally, we observe that most models output zero object rotation by default; hence, we omit rotation from evaluation to simplify the task and focus on more meaningful aspects of spatial understanding. In this paper, we evaluate all models on a representative subset of the CLEVR dataset's validation split, containing 1,500 image–scene pairs with GT annotations.

**Json Correction**    We incorporate two levels of correction in the IR3D-Bench pipeline to make the proposed pipeline robust: Rule-based correction: We apply structural validation using regular expressions and syntax normalization to fix common JSON formatting issues (e.g., missing brackets, misplaced commas, incorrect types). LLM-based correction: We further use a secondary LLM (GPT-4o in our experiments) to correct formatting errors while preserving semantics. This significantly improves the consistency of parsed outputs.

## A.2 Task prompt

**Design of Inverse Rendering Prompt**    The complete prompt used for inverse rendering is presented in Table 2. Given an input image, the vision-language model is instructed to extract scene-level geometry, material properties, and lighting parameters, and to output the results in a strict JSON format. This structured output is then parsed and used to reconstruct the 3D scene within Blender [15]. The prompt is carefully designed to ensure consistency, reproducibility, and compatibility with downstream scene synthesis workflows.

**Design of LLM Score Prompt**    The complete prompt used to elicit LLM-based evaluations of reconstructed 3D scenes is detailed in Table 3. This prompt instructs the LLM to compare a predicted scene description against a ground-truth JSON, and to assign scores across multiple dimensions of fidelity and accuracy. The evaluation process is fully self-contained, requiring the model to analyze only the provided JSON content without recourse to external assumptions. Scores are returned in a structured JSON format, with each score accompanied by a concise justification referencing specific attributes or spatial discrepancies.

**Design of Refinement Prompt**    The complete prompt used for refining the scene description based on prior outputs is presented in Table 4. This prompt instructs the LLM to revise the predicted JSON scene by leveraging both the ground-truth image and the rendered image from the current JSON, under a fixed camera setup. The objective is to produce a refined scene JSON that is visually and spatially aligned with the ground-truth reference, in terms of object layout, attributes, and relationships. LLM is required to output a valid JSON object conforming to a strict schema (the same as the inverse rendering prompt), with no additional text, ensuring consistency and interpretability for downstream evaluation.

# B More Experimental Results

## B.1 More Visual Results

We present additional qualitative comparisons in Figure 8. Among all models, Gemini-2.5 Pro [76] achieves the most faithful reconstructions in terms of geometry, spatial layout, and material appearance. Grok-3 [77] shows competitive performance in recovering fine material details, though with minor spatial inconsistencies. In contrast, models such as Mistral-3 [90], InternVL-Chat-3B [87],

Table 2: Prompt for Inverse Rendering

| 3D Inverse Rendering Prompt Specification |
|---|

| #1 | **Task Description**
Please carefully analyze the provided image, identifying all major geometric objects, their properties, and the scene's lighting setup. Your task is to extract object and lighting information and return the result strictly following the JSON format specified below. The Camera parameters are fixed and should be used as provided in the JSON structure. This JSON will be used by a Python script to reconstruct the scene in Blender. |
|---|---|
| #2 | **Output Format Requirements**
• Please output **only** a valid JSON object, without any additional explanations, comments, or code block markers (like '''json ...'''). The JSON object must adhere to the following structure: |

```
"camera": {
  // --- CRITICAL: Use these FIXED values for the camera ---
  "name": "MainCamera", "location": [0.0, -10.0, 5.0], "lens": 50.0,
  "rotation_euler": [65.0, 0.0, 0.0], // Provided in degrees
  "sensor_width": 36.0, "sensor_height": 36.0, "clip_start": 0.1, "clip_end": 100.0
  // --- Do NOT estimate camera parameters ---
},
```

```
"lighting": {
  "sun_energy": float, // Estimated sun light intensity (e.g., between 2.0 and 5.0)
  "sun_rotation_euler_degrees": [float, float, float], // Rotation angles [X, Y, Z]
  "environment_color": [float, float, float, float], // RGBA, e.g., [0.8, 0.8, 0.8, 1.0]
  "environment_strength": float // Light strength, e.g., between 1.0 and 1.5
},
```

```
"objects": [
  {
    "name": "descriptive_name_string", // e.g., "green large metal cylinder"
    "location": [float, float, float], // [X, Y, Z]
    "rotation_euler": [float, float, float], // Usually [0, 0, 0]
    "size_params": {
      // One of the following based on shape:
      // "size": float          // If 'cube'
      // "radius": float, "depth": float // If 'cylinder'
      // "radius": float        // If 'sphere'
    },
    "material": {
      "name": "MaterialNameString", // e.g., "GreenMetal"
      "base_color": [float, float, float, float], // [R, G, B, A]
      "metallic": float, // Between 0.0 and 1.0
      "roughness": float // Between 0.0 and 1.0
    }
  } // ... Potentially multiple object entries ...
]
```

| #3 | **Object Analysis Guidelines**:
• **Identification:** Find all clearly visible, primary geometric objects in the image.
• **Name (name):** Use the format `"color size material shape"`, e.g., `"blue large rubber cube"`.
• **Location (location):** Estimate the object's center position as `[X, Y, Z]`. Assume the ground is at `Z = 0`, and Z is usually half the object's height. Estimate X and Y based on the object's left-right and front-back position in the image.
• **Rotation (rotation_euler):** For CLEVR-style images, objects are usually upright. Use `[0.0, 0.0, 0.0]` unless there is visual evidence to the contrary.
• **Size Parameters (size_params):** Based on the object shape, include:
  – cube: `"size": float` (estimated edge length)
  – cylinder: `"radius": float, "depth": float` (estimated base radius and height)
  – sphere: `"radius": float` (estimated radius)
  Estimate all size values visually relative to other objects in the image.
• **Material (material):**
  – name: Generate a concise material name, e.g., `"GreenMetal"`. Reuse the same name for identical materials across objects.
  – base_color: RGBA color in format `[R, G, B, A]`, where values are between 0.0 and 1.0.
  – metallic: Use `1.0` for metallic surfaces, `0.0` for non-metal surfaces.
  – roughness: Estimate based on surface appearance — smooth/mirror-like surfaces have low roughness (near 0.0), matte/dull surfaces have high roughness (near 1.0). |
|---|---|

Table 3: Evaluation Prompt for Scoring 3D Sene Json

| 3D Scene JSON Description Evaluator Prompt | |
|---|---|
| #1 | **Role and Task** 
 You are an AI evaluator specializing in 3D scene descriptions. Your task is to compare a **Predicted JSON** scene description with a **Ground Truth (GT) JSON** and evaluate the accuracy and consistency of the predicted scene. |
| #2 | **Instructions** 
 • The Predicted JSON will be provided first. 
 • The GT JSON will follow. 
 • Focus only on the **data in the JSONs** — no external visual interpretation or assumptions. 
 • Acknowledge and account for structural differences across JSONs. 
 • Your output must be a **single valid JSON object** following the format below — no other text or explanations. |
| #3 | **Scoring Scale (Per Dimension)** 
 • 5: Excellent — Highly accurate and consistent 
 • 4: Good — Mostly accurate, minor discrepancies 
 • 3: Fair — Captures core ideas but with noticeable issues 
 • 2: Poor — Significant inaccuracies 
 • 1: Very Poor — Major incorrect aspects 
 • 0: Completely Incorrect or Missing |
| #4 | **Evaluation Dimensions** |
| 1 | **GPT4.1-JSON Object Appearance Fidelity** 
 *Focus:* Can plausible object matches be found? How accurate are the predicted attributes vs GT? 
 • Match predicted `name` (color/size/material/shape) with GT attributes. 
 • Compare predicted `material` fields (e.g., `metallic`) with GT material category. 
 • Compare predicted `size_params` to size descriptors like "small"/"large". 
 *Score reflects object match quality and attribute-level accuracy.* |
| 2 | **GPT4.1-JSON Scene Layout Accuracy** 
 *Focus:* For matched objects, how close are predicted `location` values to GT `3d_coords`? 
 *Score reflects 3D spatial alignment accuracy.* |
| 3 | **GPT4.1-JSON Overall Visual Quality & Similarity** 
 *Focus:* Holistic assessment of how well the predicted JSON matches the GT. 
 • Consider object count, attributes, locations. 
 • Identify any major inconsistencies within the predicted data. 
 *Score reflects overall scene description quality.* |
| #5 | **Expected Output Format (After receiving both JSONs)** 

 ```json { "GPT4_1_JSON_Object_Appearance_Fidelity": { "score": <integer 0-5>, "justification": "<string explanation of matching success and attribute accuracy, noting structural differences and specific examples>" }, "GPT4_1_JSON_Scene_Layout_Accuracy": { "score": <integer 0-5>, "justification": "<string qualitative assessment of the similarity between predicted locations and GT 3d_coords for matched objects>" }, "GPT4_1_JSON_Overall_Visual_Quality_and_Similarity": { "score": <integer 0-5>, "justification": "<string explanation for the overall data accuracy score>" } } ``` |

Table 4: Structured refinement prompt for 3D scene JSON correction based on GT and predicted image comparison.

| | **Refinement Prompt Based on GT and Predicted Images** |
|---|---|
| #1 | **Inputs** 
 • **GT Image**: Ground truth image of the scene (accurate reference). 
 • **Pred Image**: Rendered image from the current JSON scene. 
 • **Current JSON**: Scene description generated from the predicted image. |
| #2 | **Refinement Goals** 
 • Objective: Refine the parameters of all objects in a 3D scene JSON to closely match a provided ground truth (GT) image, under a fixed camera setup. NOTICE: The refined attributes of all objects in the refined json file should not be all the same as the input json file. 
 Goal: Achieve a refined scene JSON whose rendered image (with the fixed camera) is visually and spatially consistent with the GT image, in terms of object count, placement, size, shape, material, and inter-object relationships. |
| #3 | **Constraints** 
 • All changes must be grounded in visual evidence from GT vs Pred and metric feedback. 
 • The final output must be a valid JSON object that strictly follows the original schema. 
 • Output only the JSON object — no extra explanation, comments, or formatting. |
| #4 | **Output Format Requirements** 
 • Please output only a valid JSON object, without any additional explanations, comments, or code block markers (like json ... ). The JSON object must adhere to the following structure: 
 • Camera format: |

```
"camera": {
  // --- CRITICAL: Use these FIXED values for the camera ---
  "name": "MainCamera", "location": [0.0, -10.0, 5.0], "lens": 50.0,
  "rotation_euler": [65.0, 0.0, 0.0], // Provided in degrees
  "sensor_width": 36.0, "sensor_height": 36.0, "clip_start": 0.1, "clip_end": 100.0
  // --- Do NOT estimate camera parameters ---
},
```

• Lighting format:

```
"lighting": {
  "sun_energy": float, // Estimated sun light intensity (e.g., between 2.0 and 5.0)
  "sun_rotation_euler_degrees": [float, float, float], // Rotation angles [X, Y, Z]
  "environment_color": [float, float, float, float], // RGBA, e.g., [0.8, 0.8, 0.8, 1.0]
  "environment_strength": float // Light strength, e.g., between 1.0 and 1.5
},
```

• Objects format:

```
"objects": [
  // --- CRITICAL: Refine the parameters of each object ---
  {
    "name": "descriptive_name_string", // e.g., "green large metal cylinder"
    "location": [float, float, float], // [X, Y, Z]
    "rotation_euler": [float, float, float], // Usually [0, 0, 0]
    "size_params": {
      // One of the following based on shape:
      // "size": float            // If 'cube'
      // "radius": float, "depth": float // If 'cylinder'
      // "radius": float          // If 'sphere'
    },
    "material": {
      "name": "MaterialNameString", // e.g., "GreenMetal"
      "base_color": [float, float, float, float], // [R, G, B, A]
      "metallic": float, // Between 0.0 and 1.0
      "roughness": float // Between 0.0 and 1.0
    }
  } // ... Potentially multiple object entries ...
]
```

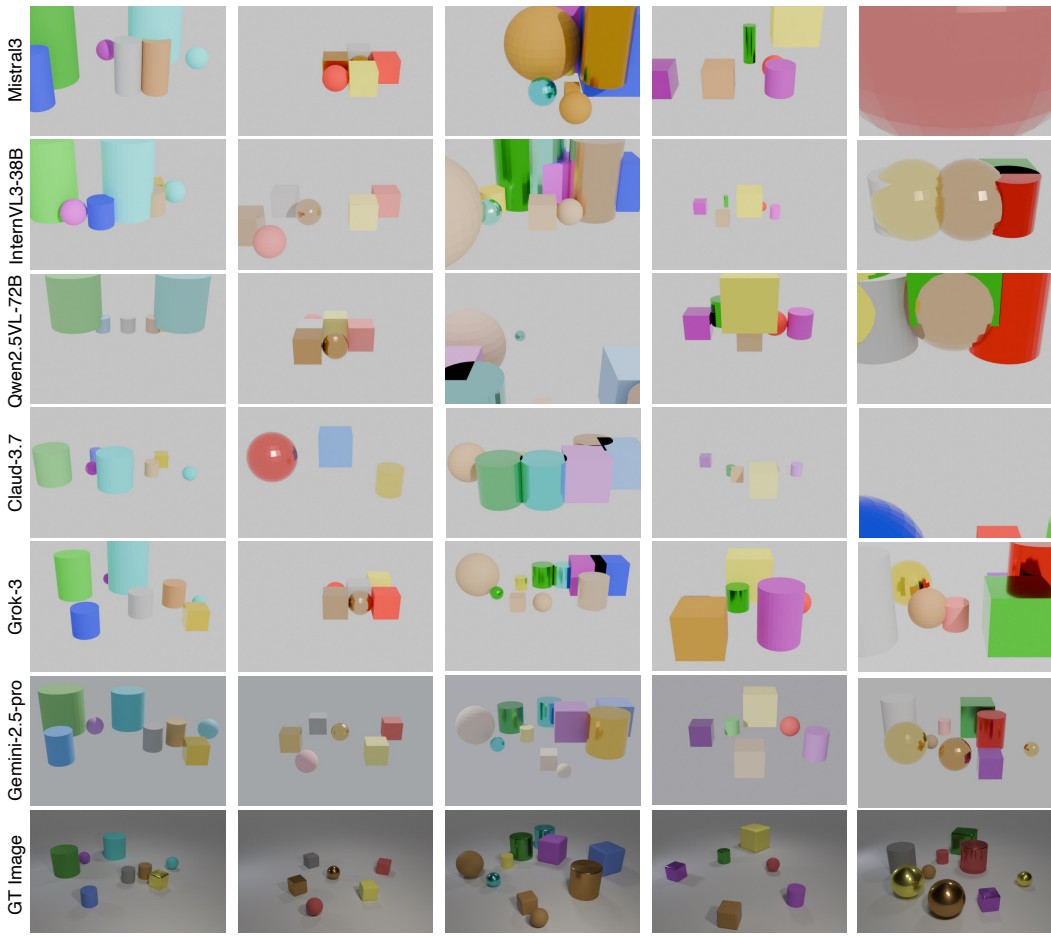

Figure 8: More Visual Results with Selected VLMs on IR3D-Bench

Qwen2-5LVL-72B [88], and Claude-3-7 [3] often exhibit noticeable errors in object positioning and material prediction, leading to spatial misalignments, incorrect relative depths, and inconsistencies in color or reflectance. These observations further highlight the strengths of Gemini-2.5 Pro in holistic scene understanding.

## B.2 Impact of Prompt Design

To quantitatively assess the contribution of different prompt components, we perform ablation studies on four key elements: structured output format, fixed camera configuration, task decomposition, and attribute specification. We use GPT-4o [89] as the VLA for conducting these experiments. As shown in Table 5, the full prompt achieves the best overall performance across almost all evaluation metrics, with a total Pixel Distance of 0.5528 and DICE score 0.11. Removing the structured format results in outputs lacking essential attributes required for rendering, thereby making it fail to render and compute downstream reconstruction metrics. This highlights the critical role of structured prompts in constraining the output space and ensuring syntactic and semantic completeness. Eliminating the fixed camera setup results in worsened spatial consistency, particularly evident in reduced shape accuracy (99.14 vs. 99.88) and a lower layout score (1.91 vs. 1.94). Without task decomposition and clarification, the model struggles with compositional reasoning, leading to a drop in object count accuracy (0.91 vs. 0.94) and Pixel Distance (0.7156 vs. 0.5528). Finally, omitting attribute guidelines significantly impacts fine-grained predictions, notably reducing material accuracy (97.59 vs. 98.66) and object-level score (2.73 vs. 2.90). These results empirically validate that each prompt component plays a critical role in guiding the VLM toward faithful and consistent scene reconstruction.

Table 5: **Quantitative results of models with various prompt designs on IR3D-Bench.** We report performance across key aspects of 3D scene reconstruction from a single image.

| Models | Layout & Localization | | | Relation | Instance Seg. | | CLIP Score | | | | | LLM Score | | |
|---|---|---|---|---|---|---|---|---|---|---|---|---|---|---|
| | Pix. Dist.↓ | Count ACC↑ | Bbox↑ | Rel. ACC↑ | IOU↑ | DICE↑ | Color↑ | Size↑ | Material↑ | Shape↑ | Overall↑ | Obj↑ | Layout↑ | Overall↑ |
| w.o. Given Format | - | - | - | - | - | - | - | - | - | - | - | - | - | - |
| w.o. Given Camera | 0.7156 | **0.94** | 0.06 | 0.24 | 0.02 | 0.03 | 96.57 | 97.58 | 98.02 | 99.59 | 93.46 | 2.69 | 1.91 | 2.38 |
| w.o. Task Analysis | 0.6718 | 0.91 | 0.27 | **0.28** | 0.06 | 0.09 | 97.01 | 97.53 | 97.79 | 99.14 | 93.65 | 2.84 | **2.05** | 2.31 |
| w.o. Attribute Guides | 0.5891 | **0.94** | 0.26 | 0.27 | **0.07** | 0.1 | **97.15** | 98.15 | 97.59 | 99.59 | 94.03 | 2.73 | 1.85 | 2.01 |
| GPT-4o | **0.5528** | **0.94** | **0.29** | 0.3 | **0.07** | **0.11** | 96.7 | **98.36** | **98.66** | **99.88** | **94.22** | **2.9** | 1.94 | **2.52** |

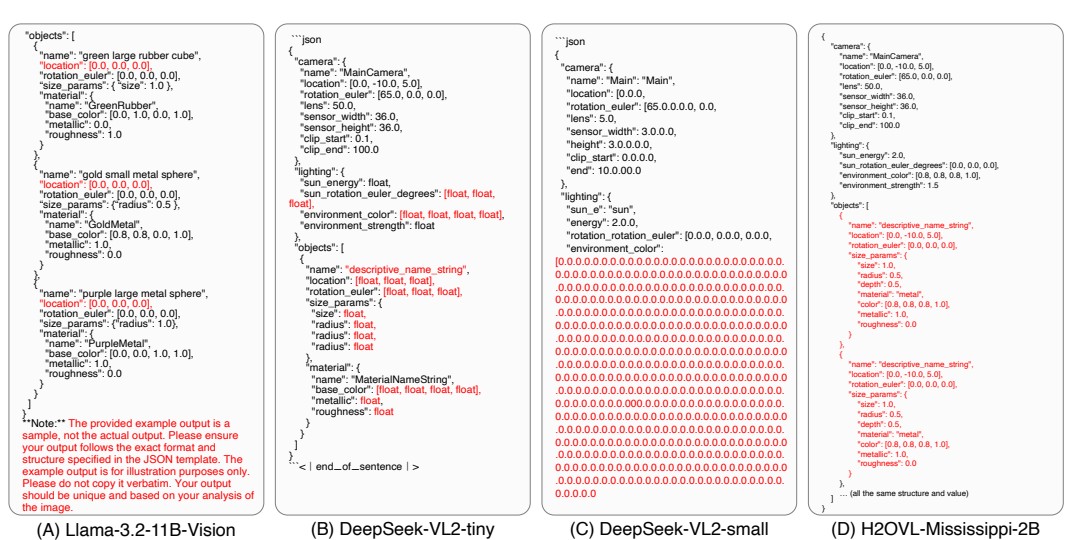

Figure 9: Failure output of selected models on IR3D-Bench

## B.3 Analysis of Failure Cases

Among the evaluated models, LLaMA-3.2-11B-Vision [80], DeepSeek-VL2 [78] (tiny and small version), and H2OVL-Mississippi-2B [81] represent clear failure cases on IR3D-Bench. Despite the general success of many models, these models consistently fail to produce valid outputs suitable for inverse rendering. As illustrated in Figure 9, LLaMA-3.2-11B-Vision repeatedly emits a static template across all test cases, with identical object attributes and all object locations fixed at $(0, 0, 0)$, rendering the outputs semantically meaningless and non-renderable. DeepSeek-VL2-tiny directly copies the JSON template, including placeholders such as [float, float, float], without generating any instance-specific values. Similarly, DeepSeek-VL2-small fails to populate essential fields, instead outputting large arrays of zeros without assigning any object-level attributes. H2OVL-Mississippi-2B generates multiple objects, but with identical and repetitive attributes across all instances, suggesting template replication without true scene interpretation. In all these cases, the lack of structurally valid and semantically grounded output prevents rendering and quantitative evaluation, highlighting the importance of model understanding and prompt grounding in 3D scene tasks.

# C Further Analysis

## C.1 Limitation

While IR3D-Bench offers a novel lens to evaluate vision-language models (VLMs) through agentic inverse rendering, several limitations remain. First, the benchmark is constructed on the CLEVR dataset, which contains synthetic scenes with clean geometry and controlled semantics. While this design enables precise evaluation, it lacks the visual richness and noise inherent in real-world data. We intentionally refrain from using real-world datasets at this stage because most models still struggle to perform reliably even under the simplified CLEVR setting. Second, our current evaluation focuses on single-view, static scene reconstruction, without considering temporal consistency or

multi-view fusion, both of which are essential for reasoning in dynamic and embodied environments. Lastly, we fix the camera intrinsics and extrinsics and omit illumination modeling. This decision is made not only to reduce task ambiguity, but also because current models already exhibit substantial limitations in reconstructing geometry and semantics under these simplified conditions. Introducing additional complexity at this stage may obscure rather than clarify the core challenges in agentic visual understanding.

## C.2 Future Extension

IR3D-Bench provides a structured setting that can support the construction of high-quality instruction-output pairs for supervised fine-tuning (SFT) or chain-of-thought (CoT) training. By leveraging successful inverse rendering examples, we can curate targeted datasets to improve VLMs' compositional reasoning and program generation capabilities. Building on this, future extensions of IR3D-Bench could extend to multi-view and dynamic scenes, and incorporate camera parameter estimation and illumination modeling. Ultimately, extending the benchmark to real-world datasets with diverse appearance, clutter, and geometry will enable comprehensive evaluation and training of VLAs in open-world, visually complex environments.

