# OpenReview forum: "IR3D-Bench: Evaluating Vision-Language Model Scene Understanding as Agentic Inverse Rendering"
_NeurIPS.cc/2025/Datasets_and_Benchmarks_Track — NeurIPS 2025 Datasets and Benchmarks Track poster_

### Official Review · Reviewer_YRGG · 2025-06-26

**Rating:** 4
**Confidence:** 4

**Summary:**

This paper introduces IR3D-Bench, a novel benchmark for evaluating the scene understanding capabilities of Vision-Language Models by requiring them to perform agentic inverse rendering: reconstructing a 3D scene from a single 2D image via executable Blender code. Rather than assessing descriptive capabilities like captioning or VQA, IR3D-Bench evaluates whether models can “understand by creating” through tool use. The benchmark integrates the CLEVR dataset, provides a structured prompt and JSON schema, and includes a comprehensive suite of evaluation metrics covering geometry, visual appearance, and semantic coherence. Extensive experiments demonstrate both promise and limitations in current model capabilities.

**Dataset Code Accessibility:**

Yes

**Dataset Code Comments:**

The dataset is uploaded to HuggingFace. Code is hosted anonymously.

**Ethical Comments:**

The benchmark only uses synthetic and public CLEVR images, eliminating concerns about consent, privacy, or real-world bias.

**Ethical Considerations:**

No, there are no or only very minor ethics concerns

**Final Justification:**

The rebuttal adequately addresses my concerns, and I will keep the current score.

**Limitations Weaknesses:**

- The benchmark depends on the CLEVR dataset which only has simple primitive shapes, which is hard to reflect realistic 3D scenes and might be biased. If a VLM is specifically fine-tuned on such data domain, it might overfit well to the benchmark but still fail to understand true 3D scenes.
- General VLMs might be hard to directly predict numbers (like the position of an object in absolute coordinate system & camera poses). In fact, human also cannot predict these numbers accurately without special tools. It remains to be discussed if predicting numbers is a proper way for benchmarking 3D understanding.
- The color palette of Figure 4 is hard to distinguish.

**Strengths Contributions:**

- The benchmark designs a clever and controlled setting to compare the scene understanding capability of VLMs, inspired by the understanding by creating idea.
- The benchmark has enough test samples to ensure the robustness of the model.
- The benchmark evaluates many different aspects of scene understanding, including localization, visual appearance, and semantics.
- The paper is well-written and clearly structured.

---

> ### Author Rebuttal · Authors · 2025-07-31
>
> We thank the reviewer for recognizing our work as “**_a clever and controlled setting_**” and for highlighting its “**_robust testing_**”, “**_comprehensive evaluation_**”, and “**_clear writing_**”.
> We hope our responses below help address the concerns and demonstrate the value and extensibility of our work:
>
> ### Q1. Gap When Applied to Real Scenes?
> IR3D-Bench remains effective when extended to real-world scenarios. We present the detailed analysis below.
>
> The main goal of IR3D-Bench is to **benchmark the spatial understanding capabilities of VLMs through reconstruction**, but not to reconstruct or generate realistic 3D scenes.
> Although CLEVR scenes consist of simple shapes, **inferring multi-level object attributes (e.g., position, color, material) for reconstruction is still non-trivial for VLMs**.
> We observe that existing VLMs still exhibit a significant performance gap compared to the ground truth on CLEVR scenes, which motivates our focus on CLEVR scenes as a practical and informative setting in the 'understanding-by-creating' manner.
>
> To show the IR3D-Bench's generalization on real scenes, we additionally **collected a real-world dataset that contains multi-object scenes that resemble those found in real-world environments** (e.g., multiple familiar, real-world objects placed on a table), with objects that are visually more complex than those in CLEVR and images captured under challenging and varied lighting conditions. Experimental results are listed below:
> | Model             | Count ACC | BBox Score | Relation ACC | Mask IOU | Mask DICE | CLIP Color | CLIP Shape | LLM Obj | LLM Layout | LLM Overall |
> |-------------------|-----------|------------|--------------|----------|-----------|------------|------------|---------|------------|-------------|
> | Gemini-2.5-pro    | **0.7000** | 0.5629     | **0.3292**   | **0.0907** | **0.1290** | 0.9998     | 0.9997     | 4.3000  | 3.3000     | 4.3000      |
> | Gemini-2.5-flash  | 0.6250    | 0.5732     | 0.2083       | 0.0513   | 0.0761    | 0.9996     | **0.9999** | 4.4000  | 3.1000     | 4.1000      |
> | GPT-4o            | **0.7000** | 0.5221     | 0.2250       | 0.0780   | 0.1225    | 0.9997     | 0.9998     | 4.3000  | 3.1000     | 4.1000      |
> | Claude-3-7-sonnet | **0.7000** | 0.6034     | 0.2313       | 0.0642   | 0.0942    | 0.9999     | 0.9996     | 4.4000  | 3.2000     | 4.2000      |
> | Claude-3-5-sonnet | 0.6333    | 0.4636     | 0.1690       | 0.0405   | 0.0706    | 0.9995     | 0.9997     | 4.6000  | 3.4000     | **4.4000**  |
> | Grok-3            | **0.7000** | **0.6448** | 0.3188       | 0.0814   | 0.1201    | **0.9999** | 0.9997     | **4.8000** | **3.4000** | **4.4000**  |
>
> **Consistent trends with the results of IR3D-Bench can be observed from the results**:
> - Gemini-2.5-pro shows the most overall balanced performance.
> - Grok-3 excels in attributes like material and color details.
>
> Fine-tuning on real-world data can further improve the model's understanding of real 3D scenes.
>
> ### Q2. How Does IR3D-Bench Predict Object Positions and Camera Poses in Absolute Coordinates?
> IR3D-Bench provides spatial grounding via structured inputs, enabling VLMs to make informed and interpretable 3D predictions rather than guessing arbitrary numbers.
>
> We agree that precisely perceiving a number, like the position of an object, can be challenging based on just one image input.
> Therefore, **IR3D-Bench does not require VLMs to guess raw numbers while we provide the spatial context**, including the 3D coordinate system and camera pose, via the formant JSON input so that VLMs can have a sense on the standard of numbers in such systems.
> Our experiments show that models tend to produce objects with incorrect scale or misaligned positions when it is absent.
> With this spatial grounding, we find VLMs are able to **make informed predictions by referencing known objects in the scene and comparing visual cues such as relative size, distance, and alignment**.
> Here is a thinking process from Gemini-2.5-Pro:
> ```vbnet
> location: [-3.0, 0.0, 1.25]
> Reasoning:
> X = -3.0 → The object is to the left of the central cube, at a distance slightly greater than the cube’s width.
> Y = 0.0 → It lies on the same forward-backward plane as the central cube.
> Z = 1.25 → The object’s height (depth) is estimated at 2.5, so its center is at 2.5 / 2 = 1.25.
>
> size_params: {"radius": 1.2, "depth": 2.5}
> Reasoning:
> depth = 2.5 → Slightly taller than the large gray cube (2.0 units).
> radius = 1.2 → Its diameter appears wider than the gray cube’s edge (2.0), so a radius of 1.2 (diameter 2.4) is reasonable.
> ```
> This shows that the predicted values are not arbitrary, but grounded in spatial understanding of the scene. We thus consider structured numerical output, when anchored to a known coordinate system, a valid and informative benchmark for 3D understanding in VLMs.
>
>
> ### Q3. Color Palette of Radar Figure
> Thank you for the suggestion, and we will improve the color palette in Figure 4 for better readability in the camera-ready version.

---

> > ### Comment · Reviewer_YRGG · 2025-08-02
> >
> > Thank you to the authors for the rebuttal. The spatial context finding is interesting, and I believe a more in-depth analysis could inspire future work. I will maintain my current score.

---

> > > ### Author Response · Authors · 2025-08-02
> > > **Appreciation for Constructive Feedback**
> > >
> > > We appreciate your positive feedback on the spatial context finding, which could inspire future work. We are also grateful to see that the revision of IR3D-bench could impact a broader audience after clarifying the in-depth analysis informed by your constructive comments.

---

### Official Review · Reviewer_tBJd · 2025-07-01

**Rating:** 6
**Confidence:** 4

**Summary:**

this work proposed a benchmark to evaluate whether VLMs truly understand scenes from only visual observations.  That is the VLMs understanding capability is evaluated on active creation rather than conventional passive understanding. The full framework follows a understand-by-creation pipeline with VLA.  A comprehensive set of metrics including geometrical accuracy, spatial relationship, appearance attributes, and overall plausibility are introduced to move the evaluation beyond metrics under traditional VLM secne understanding benchmarks.  Some intial experiments are conducted to show the limitations of current sota VLMs, highlighting the potential of this evaluation benchmark.

**Dataset Code Accessibility:**

Partly

**Ethical Considerations:**

No, there are no or only very minor ethics concerns

**Final Justification:**

For evaluating 3D VLMs capability, this work provides a solid dataset and benchmark from solely scene understanding to more active scene generation for evaluation capability. And all my concerns are well addressed during rebuttal and reviewer-author discussion phase, thus i decided to raise my rating as strong accept.

**Limitations Weaknesses:**

1. for visual correspondance evaluation, why 3D evaluation metrics like chamfer distance are not incoporated to evaluate the inverse-rendering results by blender ? it's encouraged to directly employ such metrics to evaluate VLMs spatial understanding capability.

2. the CLEVR dataset is limited at texture less 3D objects, which would be too simple for current VLMs to understand object attributes.  Blender can support more versatile textures, authors may introduce more complicated 3D datasets, or use synthesis data from some 3D objects dataset like Objaverse.

3. It's better to include a video with flying-through rendering views of ground truth scenes and inverse-rendering scenes to show the limitation of exisiting VLMs in this benchmark.

**Strengths Contributions:**

1. *What I cannot create, I do not understand.* It's generally a interesting benchmark to explore for VLMs, especially compared to convential 3D VLM evaluation benchmarks, like scene understanding (grounding, localization) benchmark built upon scannet dataset.

2. Figure.2 is well presented to show the pipeline and three types of evaluation metrics including localization, visual appearance and language-aligned semantic metrics.

3. Taking Blender engine into the pipeline through VLA generated json files makes the pipeline looks novel compared to conventional VLM evaluation pipelines, and can further support datasets beyond datas like CLEVER.

---

> ### Author Rebuttal · Authors · 2025-07-31
>
> We sincerely thank the reviewer for recognizing our work as "**_an interesting exploration for VLMs_**," "**_well presented_**," and for its "**_novel pipeline_**."
> We hope our responses below help address the reviewer's major concerns on adding a 3D evaluation metric (_Q1_) and Performance extended to complex scenes with textures? (_Q2_), and also thank you for the suggestion on further visualization (_Q3_).
>
> ### Q1. Add a 3D Evaluation Metric
> Thank you for your insightful question regarding the usage of 3D evaluation metrics. In IR3D-Bench, 2D Pixel Distance shows strong alignment with Chamfer Distance, making it a reliable proxy.
>
> We evaluate geometric alignment **via 2D Pixel Distance (col. 3 in Tab. 1) after projecting 3D points onto the image plane**. In contrast to Chamfer Distance in 3D, our metric operates in image space. Notably, we observe that **the results produced by the two metrics are closely aligned across models**, suggesting that our 2D metric serves as a reliable and practical proxy for evaluating spatial understanding in inverse rendering tasks.
> |                  | Gemini-2.5-pro | Gemini-2.5-flash | Grok-3 | Claude-3.7-Sonnet | GPT-4o |
> |------------------|----------------|------------------|--------|-------------------|--------|
> | Chamfer Dist.    | 0.3881         | 0.4075           | 0.4104 | 0.4784            | 0.4364 |
> | Pixel Dist.      | 0.3791         | 0.4291           | 0.4378 | 0.5099            | 0.5528 |
>
> Nevertheless, we fully agree that incorporating explicit 3D metrics like Chamfer Distance can be beneficial, especially as we scale to more complex scenes or explore fine-grained spatial understanding.
>
>
> ### Q2. Performance Extended to Complex Scenes With Textures?
> IR3D-Bench maintains consistent performance in complex, textured scenes, showing strong generalization. The detailed analysis is outlined below.
>
> **IR3D-Bench aims to benchmark the spatial understanding ability of VLMs via scene reconstruction**, in contrast to prior works on image-to-3D generation or reconstruction [1] [2].
> Despite the limited textures in CLEVR, **inferring multi-level attributes for each object (like position, shape, and color) remains challenging for current models**. Therefore, IR3D-Bench uses CLEVR scenes as a suitable setting to evaluate VLMs' spatial understanding in an understanding-by-creating manner.
>
> **Objaverse [3] is not suitable for IR3D-Bench** because it consists of individual objects rather than scenes, making spatial understanding tasks infeasible.
> To show the generalization of IR3D-Bench to more complicated scenes, **we collect a new dataset containing multi-object scenes that resemble those found in real-world environments, with objects that are visually more complex than those in CLEVR and images captured under challenging and varied lighting conditions,** e.g. multiple familiar, real-world objects placed on a table. The results are presented as follows:
>
> | Model             | Count ACC | BBox Score | Relation ACC | Mask IOU | Mask DICE | CLIP Color | CLIP Shape | LLM Obj | LLM Layout | LLM Overall |
> |-------------------|-----------|------------|--------------|----------|-----------|------------|------------|---------|------------|-------------|
> | Gemini-2.5-pro    | **0.7000** | 0.5629     | **0.3292**   | **0.0907** | **0.1290** | 0.9998     | 0.9997     | 4.3000  | 3.3000     | 4.3000      |
> | Gemini-2.5-flash  | 0.6250    | 0.5732     | 0.2083       | 0.0513   | 0.0761    | 0.9996     | **0.9999** | 4.4000  | 3.1000     | 4.1000      |
> | GPT-4o            | **0.7000** | 0.5221     | 0.2250       | 0.0780   | 0.1225    | 0.9997     | 0.9998     | 4.3000  | 3.1000     | 4.1000      |
> | Claude-3-7-sonnet | **0.7000** | 0.6034     | 0.2313       | 0.0642   | 0.0942    | 0.9999     | 0.9996     | 4.4000  | 3.2000     | 4.2000      |
> | Claude-3-5-sonnet | 0.6333    | 0.4636     | 0.1690       | 0.0405   | 0.0706    | 0.9995     | 0.9997     | 4.6000  | 3.4000     | **4.4000**  |
> | Grok-3            | **0.7000** | **0.6448** | 0.3188       | 0.0814   | 0.1201    | **0.9999** | 0.9997     | **4.8000** | **3.4000** | **4.4000**  |
>
> **The performance trends across models are consistent with those observed in CLEVR**, which further supports the generalizability of our pipeline to more complex and realistic settings.
> We agree that texture is an important component in inverse rendering and scene understanding, and we plan to extend IR3D-Bench to support scenes with richer textures in future work, enabling more comprehensive evaluation of VLMs in open-world 3D understanding.
>
>
> ### Q3. Video Visualization
> Thank you for the suggestion, a fly-through comparison would indeed highlight current VLM limitations.
> Unfortunately, we can't include it in the rebuttal due to link restrictions, but we will add it to the camera-ready appendix.
>
> [1] Long X, Guo Y C, Lin C, et al. Wonder3d: Single image to 3d using cross-domain diffusion. CVPR 2024.
>
> [2] Tang J, Chen Z, Chen X, et al. Lgm: Large multi-view gaussian model for high-resolution 3d content creation. ECCV 2024.
>
> [3] Deitke M, Liu R, Wallingford M, et al. Objaverse-xl: A universe of 10m+ 3d objects. NeurIPS 2023.

---

> > ### Comment · Reviewer_tBJd · 2025-08-07
> >
> > I appreciate authors' additional experiments and thoughtful rebuttal. My concerns regarding Q2 and Q3 are well addressed in the authors rebuttal, as figures and videos are not applicable during the neurips rebuttal. But i still have some additional questions about Q1.
> >
> > Though metric like 2D Pixel Distance can be view as proxy of 3D metrics like chamfer distance, they need extra projection on 2D images and only can be evaluated on limited view numbers. Why 3D metrics cannot be directly applied ? are there any obstacles to directly employ metrics like 3D metrics ?
> >
> > Here additional experiments are not expected, only a discussion is expected to be provided.

---

> > > ### Author Response · Authors · 2025-08-07
> > > **Clarifying the Applicability of 3D Metrics and Justifying the Use of 2D Pixel Distance**
> > >
> > > First,  we are grateful to see that your concerns regarding _Q2_ and _Q3_ have been fully addressed. Regarding your follow-up questions on _Q1_, specifically (1) why 3D metrics cannot be directly applied and (2) whether there are any obstacles to using such metrics, we would like to clarify that **3D metrics CAN indeed be directly applied in our pipeline, and there are NO obstacles to their use**.
> > >
> > > We suspect that some confusion may have arisen from our use of the term "proxy" in the rebuttal. In certain optimization contexts, "proxy" may imply a surrogate objective used when the primary metric is inaccessible.
> > > However, in our case, we simply mean that the **2D pixel distance and the 3D pairwise metrics essentially reflect similar underlying property**, the spatial localization of the object, just from different perspectives: 2D image space versus 3D space.
> > >
> > > This means that either metric, our 2D pixel distance or 3D metrics, can be used independently, or both can be used together for evaluation. This is further shown in the Table of our response to _Q1_, as **both metrics are valid and yield a consistent relative ranking of model performance**.
> > > Thus, we will include this evaluation with the Chamfer distance you suggested in the revision and look forward to a comprehensive evaluation that strengthens the impact of IR3D on the audience, thanks to your constructive feedback.

---

> > ### Author Response · Authors · 2025-08-08
> > **Follow-up on Discussion**
> >
> > May I know if there is any further issue? We are looking forward to fully address all your concerns timely as it would be hard for further discussion after DDL (only 24 hours later).
> >
> > We deeply value your expertise and dedication in helping us improving IR3D-Bench！

---

> > > ### Comment · Reviewer_tBJd · 2025-08-08
> > >
> > > Dear authors,
> > >
> > > Thanks for your additional clarification. My concerns are well addressed, i dont have further questions to ask. Overall, i think this work is novel and worth publication. I will raise my rating as strong accept though my review is flagged insufficient by the area chair. Good luck.
> > >
> > > Regards,

---

> > > > ### Author Response · Authors · 2025-08-08
> > > > **Appreciation for Constructive Feedback**
> > > >
> > > > Thank you for your positive feedback. We're pleased that our responses have addressed your concerns. Your insightful suggestions during the rebuttal helped us improve IR3D-Bench, making it more comprehensive and broadening its impact on the audience.

---

> ### Comment · Area_Chair_j282 · 2025-08-06
> **Please read the author’s responses**
>
> Please read the author’s responses, update review to include final justification, update rating and submit mandatory acknowledgement for author response. Thanks!

---

### Official Review · Reviewer_4M4X · 2025-07-02

**Rating:** 5
**Confidence:** 3

**Summary:**

The paper introduces IR3D-Bench, a benchmark to evaluate whether Vision-Language Models can truly understand scenes by actively creating 3D structures from images, rather than just recognizing them. This benchmark challenges Vision-Language Agents to use programming and rendering tools to reconstruct 3D scenes, focusing on tool use and generative capacity.

**Dataset Code Accessibility:**

Yes

**Ethical Considerations:**

No, there are no or only very minor ethics concerns

**Final Justification:**

The rebuttal address most of my concerns. I will keep my current score.

**Limitations Weaknesses:**

1. The paper relies on scenes from the CLEVR dataset, which contains simple objects and controlled lighting conditions. While this is beneficial for isolating specific challenges in agentic inverse rendering, it may not fully reflect the complexities encountered in real-world 3D scenes.
2. The paper discusses the challenges of generating syntactically correct scripts for scene reconstruction, which is an important aspect of the agentic inverse rendering task. However, even minor errors in scripting can cause the model to fail entirely. This strict requirement may limit the model's generalization ability in more dynamic or less structured environments.

**Strengths Contributions:**

1. The paper is well-written with clearly articulated motivation.
2. Unlike traditional benchmarks that focus on descriptive tasks such as captioning or VQA, IR3D-Bench pushes the boundaries by testing models on understanding-by-creating through agentic inverse rendering. This shift from passive recognition to active creation provides a deeper test of a model's true understanding of scenes, offering a more rigorous assessment of its agentic generative capabilities.

---

> ### Author Rebuttal · Authors · 2025-07-31
>
> We sincerely thank the reviewer for acknowledging our work as "**_well written_**" with a "**_clearly articulated motivation_**," and for recognizing that it "**_pushes the boundaries of traditional benchmarks_**."
> We hope our responses below help address two major concerns and demonstrate generalization ability and the robustness of our work:
>
> ### Q1. Applicability to Real-World 3D Scenes
> IR3D-Bench is applicable to real-world 3D scenes. The detailed analysis is as follows.
>
> The goal of IR3D-Bench is to **benchmark the spatial understanding capabilities of VLMs through reconstruction**, rather than to tackle the challenge of reconstructing complex real-world scenes.
> While the objects in CLEVR are relatively simple, we observe that existing VLMs still exhibit a significant performance gap compared to the ground truth on CLEVR scenes, which **indicates that inferring multi-level object attributes for reconstruction remains a non-trivial task for VLMs**. For this reason, we chose CLEVR as a controlled starting point to evaluate and benchmark the spatial reasoning abilities of VLMs effectively.
>
> To show the applicability to real-world 3D scenes, **we evaluate a newly collected set of real-world 3D scenes**, which contains multi-object scenes that resemble those found in real-world environments, with objects that are visually more complex than those in CLEVR and images captured under challenging and varied lighting conditions (e.g. multiple familiar, real-world objects placed on a table). The results are shown below.
> | Model             | Count ACC | BBox Score | Relation ACC | Mask IOU | Mask DICE | CLIP Color | CLIP Shape | LLM Obj | LLM Layout | LLM Overall |
> |-------------------|-----------|------------|--------------|----------|-----------|------------|------------|---------|------------|-------------|
> | Gemini-2.5-pro    | **0.7000** | 0.5629     | **0.3292**   | **0.0907** | **0.1290** | 0.9998     | 0.9997     | 4.3000  | 3.3000     | 4.3000      |
> | Gemini-2.5-flash  | 0.6250    | 0.5732     | 0.2083       | 0.0513   | 0.0761    | 0.9996     | **0.9999** | 4.4000  | 3.1000     | 4.1000      |
> | GPT-4o            | **0.7000** | 0.5221     | 0.2250       | 0.0780   | 0.1225    | 0.9997     | 0.9998     | 4.3000  | 3.1000     | 4.1000      |
> | Claude-3-7-sonnet | **0.7000** | 0.6034     | 0.2313       | 0.0642   | 0.0942    | 0.9999     | 0.9996     | 4.4000  | 3.2000     | 4.2000      |
> | Claude-3-5-sonnet | 0.6333    | 0.4636     | 0.1690       | 0.0405   | 0.0706    | 0.9995     | 0.9997     | 4.6000  | 3.4000     | **4.4000**  |
> | Grok-3            | **0.7000** | **0.6448** | 0.3188       | 0.0814   | 0.1201    | **0.9999** | 0.9997     | **4.8000** | **3.4000** | **4.4000**  |
>
> **Consistent trends with the CLEVR-based IR3D-Bench can be observed in these results**: (a) Gemini-2.5-pro shows the most overall balanced performance. (b) Grok-3 excels in attributes like material and color details.
> These results further validate the effectiveness of IR3D-Bench, while also **demonstrating the feasibility of extending the benchmark to more complex scenes**.
>
> ### Q2. Is IR3D-Bench Sensitive to Script Errors?
>
> IR3D-Bench is robust to script errors and and generalizes well in real-world 3D environments.
>
> **(i) Sensitivity to Script Errors?**  We agree that directly using the model's raw JSON output can lead to failures in the inverse rendering process due to even minor syntactic errors, as we observed in our experiments, even recent closed-source models such as Gemini-2.5-Pro and Grok-3 occasionally produced malformed JSON outputs. To address this, IR3D-Bench pipeline **incorporates two levels of automatic error correction**:
> 1. Rule-based refinement: regular expressions and structural checks are utilized to correct common syntax issues (e.g., missing brackets, incorrect value types, or misplaced commas).
> 2. LLM-based refinement: a secondary LLM (GPT-4o in our experiments) is prompted to refine the JSON strictly in terms of formatting, without altering the content.
>
> With the error correction, IR3D-Bench is **highly robust in CLEVR-style scenes and almost never fails during inverse rendering**, which lays a solid foundation for future extensions to advance reasoning and more complex, realistic environments.
>
> **(ii) Will this affect the performance in real scenes?**
> At present, IR3D-Bench is **primarily intended to assess VLMs' spatial understanding via inverse rendering**, and is not designed to cover the reconstruction or generation of highly complex or dynamic scenes.
> While CLEVR is relatively simple and structured, our experiments show that existing VLMs still fall short of ground truth performance, indicating room for improvement even in this setting.
> To further show the generalization of IR3D-Bench, **evaluation on a real-world dataset collected by us (shown in _Q1_)** has shown that IR3D-Bench is capable of benchmarking the spatial understanding abilities of VLMs in an understanding-by-creating manner under realistic scenes.
> We acknowledge the limitation and plan to extend IR3D-Bench to more dynamic and less structured environments in future work.

---

> > ### Comment · Reviewer_4M4X · 2025-08-04
> >
> > Thank you for the detailed responses, which address most of my concerns. I will maintain my current score.

---

> > ### Author Response · Authors · 2025-08-04
> > **Appreciation for Constructive Feedback**
> >
> > We’re pleased that our rebuttal has addressed most of your concerns. We sincerely appreciate your insightful comments, which will help us further enhance the clarity and depth of our work in the revision.

---

### Official Review · Reviewer_3W3o · 2025-07-03

**Rating:** 5
**Confidence:** 2

**Summary:**

The authors propose a novel benchmark to evaluate scene understanding of VLMs through a novel paradigm, understanding by creating. Instead of recognition and conversational widely used in previous 3D VQA and 3D visual grounding tasks, a novel strategy that recreating the 3D scene according to the understanding is introduced to evaluate that how well does the model understand 3D scene. Specifically, VLMs need to predict attributes of objects given the raw image and corresponding task prompt, and such estimated attributes are used to re-render a reconstructed image via the python script and graphic engine. Then, several metrics aim to evaluate reconstruction quality comprehensively are introduced.

**Additional Feedback:**

+ L. 4 - L.7, several short sentences would be better for understanding.
+ The submission integrates CLEVR dataset for benchmarking object-centric 3D reconstruction and spatial reasoning in a controlled setting. However, CLEVR dataset is relatively simple and unrealistic. Is the pipeline reproducible for other datasets or re-render dataset from object dataset, e.g. Objectverse.
+ L. 292, what is the prompt of refinement?

**Dataset Code Accessibility:**

Partly

**Dataset Code Comments:**

The CLEVR dataset and code are available. However, the website of benchmark lacks detailed document.

**Ethical Considerations:**

No, there are no or only very minor ethics concerns

**Final Justification:**

All of my concerns are resolved. Therefore, I will increase my rating.

**Limitations Weaknesses:**

+ As point out by the authors in L.49, there are some errors in the process of creating. Besides, the predefined JSON format also influences the performance of understanding and reconstruction. For example, the authors define the basecolor as a 3D vector, but the basecolor of most real objects is a texture. The definition of shape also is limited for read complex objects. This assumption only works for simple dataset, like CLEVR dataset. Reducing extra errors in the creating process is crucial to unbiased evaluate the VLM's understanding of the scene.

**Strengths Contributions:**

+ A novel high-level view for evaluating scene understanding of VLMs with reconstruction via programming and rendering tools.
+ Comprehensive metrics are able to evaluate the performance. Besides, such paradigm is scalable for more metrics, e.g. comparison of object poses.
+ Extensive experiments on scene understanding and ablation study for 20 VLMs, which is a nice reference for future improvements of VLMs.

---

> ### Author Rebuttal · Authors · 2025-07-31
>
> We thank the reviewer for recognizing our work as "**_a novel high-level view_**", "**_a comprehensive evaluation_**", and "**_a nice reference for future improvements_**".
> We hope our responses below help address the reviewer's major concerns about whether predefined JSON definitions cause limitations (*_Q1_*) and reproducible for complex datasets (*_Q3_*), and also thank you for the suggestion on writing *_Q2_* and *_Q4_*.
>
> ### Q1. Do Predefined JSON Definitions Cause Limitations?
> IR3D-Bench remains robust to scripting errors and generalizes effectively to real-world scenes, due to its dual correction and abstraction-based design.
>
> **(i) Scripting Errors?** It impacts very little as we incorporate **two levels of correction in the IR3D-Bench pipeline** to address make the proposed pipeline robust:
> - Rule-based correction: We apply structural validation using regular expressions and syntax normalization to fix common JSON formatting issues (e.g., missing brackets, misplaced commas, incorrect types).
> - LLM-based correction: We further use a secondary LLM (GPT-4o in our experiments) to correct formatting errors while preserving semantics. This significantly improves the consistency of parsed outputs.
>
> Empirically, we find that inverse rendering almost never fails after these steps on CLEVR-style scenes.
>
> **(ii) Performance when Extended to Real-World Scenes?**
> The primary goal of IR3D-Bench is to **evaluate VLM understanding through reconstruction**, not perfect visual fidelity. While real-world scenes inevitably involve information loss when mapped to a simplified Blender-compatible schema (e.g., textures, lighting, geometry), we argue that this abstraction is acceptable for benchmarking structured understanding.
> Notably, **even in synthetic settings, where perfect reconstruction is possible, current models still struggle (see Table 1), highlighting the task’s inherent difficulty**. This justifies our choice to first focus on a controlled, tractable setting where semantics and structure are fully known.
> Moreover, we have extended IR3D-Bench to real-world scenarios, and find the pipeline remains applicable and effective in evaluating model performance under more complex conditions (see **_Q3_** for details).
>
>
> ### Q2. Abstract Clarity
> Thank you for the suggestion, and we will revise the abstract with short sentences for better readability.
>
> ### Q3. Being Reproducible for Complex Datasets
>
> IR3D-Bench generalizes well to complex real-world data, showing its effectiveness beyond synthetic benchmarks.
>
> First of all, IR3D-Bench aims to **benchmark the spatial understanding capabilities of VLMs through scene reconstruction**, rather than push the boundaries of VLMs in reconstructing complex real-world scenes.
> While CLEVR is relatively simple, predicting multi-level object attributes for accurate reconstruction remains a challenging task for VLMs, highlighting the difficulty of achieving spatial understanding through a reconstruction-based manner.
> For this reason, we chose CLEVR as a controlled starting point to evaluate and benchmark the spatial reasoning abilities of VLMs effectively.
> This aim we built also explains **why we didn't choose datasets like Objaverse** in IR3D-Bench as it consists of simply individual objects rather than scenes [1], which **disobeys our motivation to benchmark scene-wise spatial understanding**.
>
> Therefore, to better demonstrate IR3D-Bench's applicability to realistic data, **we collected a new dataset that better reflects the complexity of real-world**: it contains multi-object scenes that resemble those found in real-world environments (e.g. multiple familiar, real-world objects placed on a table), with objects that are visually more complex than those in CLEVR and images captured under challenging and varied lighting conditions. We applied the IR3D-Bench pipeline to this dataset and report the following results:
> | Model             | Count ACC | BBox Score | Relation ACC | Mask IOU | Mask DICE | CLIP Color | CLIP Shape | LLM Obj | LLM Layout | LLM Overall |
> |-------------------|-----------|------------|--------------|----------|-----------|------------|------------|---------|------------|-------------|
> | Gemini-2.5-pro    | **0.7000** | 0.5629     | **0.3292**   | **0.0907** | **0.1290** | 0.9998     | 0.9997     | 4.3000  | 3.3000     | 4.3000      |
> | Gemini-2.5-flash  | 0.6250    | 0.5732     | 0.2083       | 0.0513   | 0.0761    | 0.9996     | **0.9999** | 4.4000  | 3.1000     | 4.1000      |
> | GPT-4o            | **0.7000** | 0.5221     | 0.2250       | 0.0780   | 0.1225    | 0.9997     | 0.9998     | 4.3000  | 3.1000     | 4.1000      |
> | Claude-3-7-sonnet | **0.7000** | 0.6034     | 0.2313       | 0.0642   | 0.0942    | 0.9999     | 0.9996     | 4.4000  | 3.2000     | 4.2000      |
> | Claude-3-5-sonnet | 0.6333    | 0.4636     | 0.1690       | 0.0405   | 0.0706    | 0.9995     | 0.9997     | 4.6000  | 3.4000     | **4.4000**  |
> | Grok-3            | **0.7000** | **0.6448** | 0.3188       | 0.0814   | 0.1201    | **0.9999** | 0.9997     | **4.8000** | **3.4000** | **4.4000**  |
>
> The performance trends observed in this real-world dataset are consistent with those from IR3D-Bench, **confirming that our pipeline generalizes well in real-world scenes**.
>
> ### Q4. Prompt of Refinement
> The refinement prompt is available in the Tab. 3 of the supplementary material. We will add an explicit reference in the final version for clarity.
>
> [1] Deitke M, Liu R, Wallingford M, et al. Objaverse-xl: A universe of 10m+ 3d objects. NeurIPS 2023.

---

> > ### Comment · Reviewer_3W3o · 2025-08-04
> >
> > Thanks for your detailed responses. All of my concerns are resolved. As clarified by authors, the impact of scripts error is very little. Besides, the experiments in the main paper and new experiment on real-data dataset exhibit generalized evaluation performance on scene understanding. Therefore, I will increase my rating.

---

> > > ### Author Response · Authors · 2025-08-04
> > > **Appreciation for Constructive Feedback**
> > >
> > > Thank you for your positive feedback and for increasing your rating. We are glad our rebuttal has addressed your concerns, and we appreciate your recognition of our work's generalization. Your constructive suggestions will help us further improve the IR3D-Bench revision and make its analysis more comprehensive and thorough.

---

### Note · Authors · 2025-08-12

Dear PCs, SACs, ACs, and Reviewers,

Thank you for your valuable feedback. We are deeply encouraged by the consistently positive rating from all reviewers, as well as their recognition of the novelty and motivation behind IR3D. Specific highlights include descriptions such as “a novel high-level view for evaluating scene understanding via reconstruction” (Reviewer 3W3o), “pushes the boundaries by testing models through agentic inverse rendering” (Reviewer 4M4X), and “an interesting benchmark compared to conventional 3D VLM evaluations” (Reviewer tBJd). The work has also been acknowledged as a “comprehensive evaluation” (Reviewer 3W3o, YRGG).

During the rebuttal phase, our primary focus was on addressing the central concern: **applicability to real-world scenes**. To this end, we collected and conducted experiments on a new dataset that better reflects the complexity of real environments, thereby strengthening the real-world relevance of IR3D-Bench. This response was well-received and positively acknowledged by Reviewers 3W3o, 4M4X, tBJd, and YRGG, and we believe it significantly broadens the impact and applicability of our work.

We have also thoroughly addressed several additional points, all of which have been resolved through our responses:
- **Robustness to JSON errors**: In response to concerns raised by Reviewers 3W3o and 4M4X, we provided a detailed explanation of the two-level correction mechanism in the IR3D-Bench pipeline, which effectively mitigates JSON formatting errors.
- **Additional 3D metrics**: We clarified (in response to Reviewer tBJd) that 3D metrics can be directly integrated into our evaluation framework, and that both 2D and 3D metrics yield consistent performance rankings, validating their use.
- **Position and pose prediction**: We emphasized (addressing Reviewer YRGG) that IR3D-Bench does not require VLMs to predict raw numerical values, but instead allows models to make informed predictions by leveraging known objects and visual context.

We are pleased to see that all reviewers’ concerns have been addressed. By incorporating constructive feedback, the idea of IR3D-Bench has become more comprehensive and can impact a broader audience. We sincerely appreciate the time, effort, and insightful comments from all reviewers.

Best regards,
Authors of Submission 226

---

### Decision · Program_Chairs · 2025-09-18

**Decision:**

Accept (poster)

**Comment:**

**Rating Summary:** Reviewers gave scores ranging from 4 (Borderline Accept) to 6 (Strong Accept), with all concerns addressed post-rebuttal.

### Strengths
- **Novel Benchmark:** Introduces IR3D-Bench, shifting VLM evaluation from passive recognition to *agentic inverse rendering*—a novel and meaningful direction.
- **Comprehensive Metrics:** Evaluates geometry, appearance, and semantic coherence with structured protocols.
- **Thorough Experiments:** Tested across 20 VLMs; authors additionally included experiments on a new real-world dataset.
- **Robust Pipeline:** Addresses potential tool/script failures with rule-based and LLM-based error correction.
- **Community Value:** Publicly available dataset and code support reproducibility and further research.

### Verdict
This is a **well-motivated, well-executed benchmark** with strong generalization and practical value. The proposed "understanding-by-creating" paradigm fills a critical gap in VLM evaluation. **Recommend Acceptance.**